# Critical Parameters in an Enzymatic Way to Obtain the Unsweet Lactose-Free Milk Using Catalase and Glucose Oxidase Co-Encapsulated into Hydrogel with Chemical Cross-Linking

**DOI:** 10.3390/foods12010113

**Published:** 2022-12-26

**Authors:** Katarzyna Czyzewska, Anna Trusek

**Affiliations:** Group of Micro, Nano and Bioengineering, Wroclaw University of Science and Technology, Wybrzeze Wyspianskiego 27, 50-370 Wroclaw, Poland

**Keywords:** unsweet lactose-free milk, low-temperature biocatalysis, glucose oxidase, catalase, co-encapsulation, one-pot processes, chemical cross-linking of the hydrogel, reaction mixture aeration

## Abstract

The presented work involves obtaining and characterising a two-enzymatic one-pot bioreactor, including encapsulated (co-immobilised) glucose oxidase and catalase. The enzymatic capsules were applied to produce unsweet, lactose-free milk during low-temperature catalysis. Furthermore, operational conditions, like pH and aeration, were selected in the paper, which sorts out discrepancies in literature reports. All experiments were carried out at 12 °C, corresponding to milk storage and transportation temperature. Preliminary studies (for reasons of analytical accuracy) were carried out in a buffer (pH, concentration of sugars mimicking conditions in the lactose-free milk, the initial glucose concentration 27.5 g/L) verified by processes carried out in milk in the final stage of the study. The presented results showed the need for regulating pH and the aeration of the reaction mixture in the continuous mode during the process. The procedure of co-immobilisation was performed in an alginate matrix with the cross-linking of glutaraldehyde or carbodiimide while carbodiimide showed better enzymes retention inside alginate capsules. Co-encapsulated enzymes could be used for nine cycles, preserving finally about 40% of the initial activity.

## 1. Introduction

Lactose intolerance is recognized as a global problem, ranging from 57 to 65% of the population [1]. Therefore, the demand for lactose-free products is still growing in dessert products and dry cuisines [2,3]. Lactose-free products obtained with enzymatic lactose hydrolysis exhibit a higher sweetness than traditional substitutes [4], as the products of lactose bioconversion (glucose and galactose) have a total sweetness index higher than lactose. 

The most common strategies related to the reduction of sweetness from lactose-free milk are associated with the limiting degree of lactose hydrolysis to 80–90% [5], lactose decomposition based on lactic acid microbial fermentation (for ripened cheeses), or membrane techniques (ultra-, nanofiltration) [2,3,6]. Glucose decomposition from foodstuffs is also possible in an enzymatic way that engages glucose oxidase [7,8,9].

Glucose oxidase (GOX) is one of the most popular food enzymes. It catalyzes glucose oxidation to hydrogen peroxide and gluconic acid and is used to improve food shelf life and flavor [10]. In most cases, glucose oxidase frequently works with catalase (CAT). Their cooperation is classified as a two-enzymatic cascade [11]. In this case, catalase decomposes hydrogen peroxide, which acts as an inhibitor relative to glucose oxidase. On the other hand, oxygen generated by catalase is consumed by glucose oxidase and utilized as a substrate during glucose oxidation—Equations (1) and (2) [12].
(1)$$glucose+O_{2}\overset{glucose\;oxidase}{\to }gluconic\;acid+H_{2}O_{2}$$
(2)$$H_{2}O_{2}\overset{catalase}{\to }O_{2}+H_{2}O$$

Today, industrial biocatalysis is based on immobilized enzymes. Enzyme co-immobilization can be performed by (1) random co-immobilization, including encapsulation, attachment to the surface, and carrier-free approach, as well as by (2) positional co-immobilization by attachment to the surface or (3) compartmentalization, which can be realized with encapsulation or attachment to the surface [13]. In the case of glucose oxidase and catalase co-immobilization the strategies of covalent immobilization into a hydrogel based on sulfonamide chemistry [14], in situ immobilization in a hybrid interpenetrating polymer consisting of alginate, polyacrylamide, and hydroxyapatite [15], covalent immobilization in magnesium silicate [16], sol-gel process on silica inverse opals [17], catalytic membrane creation based on ultrafiltration membrane and polydopamine coating [18], an anion-exchange membrane of low-density polyethylene grafted with 4-vinylpiridine [19] as well as physical immobilization in porous magnetic chitosan microspheres [20] were proposed. Food bioconversion, independently from native or immobilized scenarios, must be safe for the consumer and allow the preservation of nutritional value (e.g., due to improving the shelf life of foods) [21]. These requirements are achieved using selected enzymes isolated from microorganisms classified as GRAS [22], selecting natural carriers for immobilization [21], and low-temperature biocatalysis [23,24]. The latter option is not standard, primarily due to the extension of the bioconversion time.

The presented work focuses on utilising the alginate matrix with additional cross-linking with glutaraldehyde (GA) or carbodiimide (EDC, N-(3-Dimethylaminopropyl)-N’-ethylcarbodiimide). During cross-linking, the free hydroxyl and carboxyl groups of the alginate backbone are engaged. Glutaraldehyde cross-linking is based on an acid-catalysed acetalisation reaction, after which the acetal-linked network hydrogel is formed [25]. Carbodiimide interacts with carboxyl groups of alginate and forms an O-acylisurea intermediate that reacts with amino groups of the enzyme [26]. The literature reports describe individual immobilization of glucose oxidase and catalase based on the chemical cross-linking with glutaraldehyde or carbodiimide. This strategy was suggested as an efficient way of effective enzyme binding to the support [27,28,29,30,31,32,33].

The presented work is related to low-temperature biocatalysis based on the cascade of glucose oxidase isolated from *Aspergillus niger* and catalase obtained from psychrotolerant bacteria of the *Serratia* genus. Following the low-temperature biocatalysis trends related to milk processing, we propose carrying out the glucose oxidation at 12 °C. Until now, the dairy processes catalyzed by glucose oxidase from *Aspergillus niger* were carried out at >25 °C [34,35,36]. One-pot processes with glucose oxidase and catalase are known [15,37]. However, the connection of glucose oxidase from *Aspergillus niger* and catalase from *Serratia* sp. is a new proposal, undescribed in literature reports. Moreover, the approach to the cross-linking issue also seems interesting, especially in the co-encapsulation of glucose oxidase and catalase in the alginate matrix. The available reports indicated the co-immobilization of glucose oxidase and catalase with glutaraldehyde on cotton [38] or platinum [39]. In the case of carbodiimide, results with activated carbon [40] and CM-Cellulose beads [41] can be found. Furthermore, the one-pot process with glucose oxidase and catalase acting in cascade mode is quite complex because of the necessity of pH regulation, aeration, and product inhibition. Additionally, reports from the literature are not consistent with the issue of the maintenance of medium properties, especially in the case of the type of aeration. 

Thus, with an innovation encapsulation strategy, we propose low-temperature biocatalysis acting in the cascade mode to reduce lactose-free milk’s sweetness. The obtained milk after the almost complete conversion of glucose is only slightly sweeter than raw milk (lactose has a relative sweetness of 25, galactose 32, and glucose 75). Moreover, we systematize the information related to the required operation conditions during the process of glucose conversion. 

## 2. Materials and Methods

### 2.1. Materials

Glucose oxidase from *Aspergillus niger* (G7141–50 KU) was purchased from Sigma-Aldrich (Munich, Germany). The recombinant catalase preparation isolated from psychrotolerant bacteria *Serratia* sp. (65,000 U/mL, purity > 99%) was obtained from Swissaustral (Athens, Georgia, USA) and β-galactosidase (NOLA™ Fit 5500) from Chr. HANSEN (Hørsholm, Denmark). Sodium alginate, HEPES (N-(2-Hydroxyethyl)piperazine-N’-(2-ethanesulfonic acid)) buffer (H3375), N-(3-Dimethylaminopropyl)-N’-ethylcarbodiimide (EDC), and glutaraldehyde (25% in H_2_O) (GA) were obtained from Sigma-Aldrich (Munich, Germany). The analytical glucose test was purchased by Biomaxima (Lublin, Poland). The other reagents were purchased from Avantor Performance Materials Poland S.A. (Gliwice, Poland).

### 2.2. Methods

#### 2.2.1. pH Regulation

To prevent the pH lowering during enzymatic glucose oxidation, two methods were proposed: the use of CaCO_3_ and the 1 M NaOH solution. In the first case, CaCO_3_ at concentrations of 1–4 g/L was added to the reaction mixture before the start of biocatalysis. The second approach tested the continuous dosing of 1 M NaOH by titrator. The effectiveness of both methods was evaluated based on the glucose conversion yield tested with the DNS (3,5-Dinitrosalicylic acid) method [42]. A sample of 0.5 mL (previously centrifugated at 6000 rpm for three minutes) was mixed with 1.5 mL of DNS reagent. After mixing, the solution was incubated for 5 min at 100 °C. Next, the sample was cooled for 25 min, and 8 mL of distilled water was added. Finally, a spectrophotometric measurement (Shimadzu UV-1800, Kyoto, Japan) at 550 nm was performed.

#### 2.2.2. Aeration

The aeration of the reaction mixture was performed in three different ways: by mixing (250 rpm); by adding the catalase, which in the presence of H_2_O_2_ generates oxygen; and by exogenous aeration with compressed air. Using an air flow meter, the air flow was determined on 3 L/min. The effectiveness of the proposed method was evaluated based on the glucose conversion yield tested with the DNS method. 

#### 2.2.3. The Kinetic Characteristics of the Cascade with Glucose Oxidase and Catalase in the Native Form

Glucose decomposition was carried out in plastic tubes at 10 mL. The reaction mixture consisted of the glucose solution prepared in 0.1 M HEPES buffer pH 6.6 at 27.5 g/L. Enzymes concentrations were in the range of 0.3–8.0 g/L and 0.2–2.4 g/L for glucose oxidase and catalase, respectively. The final volume of the reaction mixture was 1.5 mL and 20 mL for non-aeration and aeration conditions, respectively. The reaction was carried out at 12 °C. To prevent pH lowering, the CaCO_3_ was added at a concentration of 4 g/L. The glucose decomposition was performed under aeration conditions (compressed air) or without aeration, mixing at a speed of 250 rpm. The moment of the addition of enzymes to the substrate solution was recognized as the beginning of biocatalysis. The progress of the reaction was monitored by determining the glucose concentration according to the spectrophotometric method at 550 nm with DNS reagent. 

#### 2.2.4. Co-Encapsulation of Glucose Oxidase and Catalase 

The co-encapsulation of glucose oxidase and catalase was performed in the sodium alginate matrix with cross-linking with glutaraldehyde or carbodiimide. The sodium alginate at concentration 1.5% (*w*/*v*) was dissolved in 0.1 M HEPES buffer pH 6.6. Then, the mixture of enzymes was added in the concentration of 8 g/L and 1.3 g/L as well as 16 g/L and 2.6 g/L for glucose oxidase and catalase, respectively. After mixing, the solution was instilled with the Pasteur pipette into the cross-linking bath—CaCl_2_ at a concentration of 15% (*w*/*v*) prepared in 0.1 M HEPES buffer pH 6.6, which contains the enzymes at the same concentrations as in alginate mixture. The cross-linking lasted 2 h at 4 °C. After this time, the capsules were separated and washed slightly with distilled water to remove the residues of the CaCl_2_ mixture. In the next step, the cross-linking compounds (glutaraldehyde or carbodiimide) were used. In this case, capsules were incubated in the solution of 31.7 g/L glutaraldehyde prepared in 0.1 M PBS buffer (phosphate buffer) pH 7.0 or in the solution of 0.413 g/L carbodiimide dissolved in 0.05 M MES buffer (2-(N-Morpholino)ethanesulfonic acid). After 30 min, the capsules were separated and washed with distilled water. Then, they were incubated for 30 min in distilled water to remove the excess unbound cross-linking agents. Finally, the capsules were stored in 0.1 M HEPES buffer pH 6.6 at 4 °C.

#### 2.2.5. The Activity of Co-Encapsulated Glucose Oxidase and Catalase

The glucose decomposition was carried out in a thermostatic stirred-tank reactor with a total volume of 50 mL at 12 °C and a stirring velocity of 250 rpm (magnetic stirrer Thermo Scientific POLY15, Shanghai, China). The total volume of the reaction mixture (volume of capsules and substrate) was 18 mL, and the volume ratio of capsules to substrate solution was determined as 0.2. The enzymes’ concentrations in the reaction mixture were 1.33 g/L and 0.22 g/L for glucose oxidase and catalase, respectively. The glucose solution at a concentration of 9.33–95.43 g/L was prepared in a 0.1 M HEPES buffer, pH 6.6. The moment of the addition of the capsules to the substrate solution was recognized as the beginning of glucose decomposition. The progress of the reaction was determined spectrophotometrically at 550 nm, based on the DNS reagent. Experiments were carried out under aeration conditions: air flow 3 L/min.

#### 2.2.6. The Reusing of Co-Encapsulated Glucose Oxidase and Catalase 

The reusing opportunity of co-encapsulated glucose oxidase and catalase was determined in cycles. Each cycle corresponded to almost 97% glucose bioconversion by co-encapsulated preparation. The reaction mixture consisted of 27.5 g/L glucose solution prepared in 0.1 M HEPES buffer and alginate capsules with enzymes concentration of 1.33 and 2.76 g/L as well as 0.22 and 0.45 g/L for glucose oxidase and catalase, respectively. Glucose decomposition was carried out at 12 °C. The total volume of the reaction mixture was 18 mL, and the ratio of capsules to substrate solution was determined as 0.2. Experiments were carried out under aeration conditions.

#### 2.2.7. Glucose Oxidation in Lactose-Free Milk Catalyzed by Co-Encapsulated Glucose Oxidase and Catalase

Glucose oxidation was performed in a natural medium, lactose-free milk obtained in an enzymatic approach. In the experiment, cow milk UHT 0% fat (Laciate, Poland) was utilized. In the first step, the milk was pretreated with β-galactosidase (NOLA™ Fit 5500) to induce lactose decomposition according to the procedure described previously by authors [8]. After the reaction, the glucose concentration was determined with the Biomaxima analytical test. In this case, a 10 µL aliquot of the sample (milk) was added to 1 mL of analytical reagent. After mixing, the solution was incubated for 5 min at 37 °C. Next, the spectrophotometric measurement (Shimadzu UV-1800, Japan) at 500 nm was performed. 

Next, enzymatically obtained glucose was converted by co-encapsulated glucose oxidase and catalase. Carbodiimide was used as a cross-linking agent. The total volume of the reaction mixture (volume of capsules and substrate) was 18 mL, and the volume ratio of capsules to substrate solution was determined as 0.2. The enzymes concentrations in the reaction mixture were 2.76 g/L and 0.43 g/L for glucose oxidase and catalase, respectively. The lactose-free milk contained glucose at a concentration of 26.3 g/L, and the pH was 6.7. The CaCO_3_ (added as a solid) or 1 M NaOH equalized the decrease in pH during the reaction. Glucose decomposition was carried out at 12 °C. The moment of the addition of the capsules to the lactose-free milk was recognized as the beginning of glucose decomposition. The progress of the reaction was determined spectrophotometrically at 500 nm, based on the Biomaxima analytical glucose test. Experiments were carried out under aeration conditions (air flow 3 L/min). The mechanical foam blower (craft product) was applied to avoid the excessive foaming of milk during aeration.

## 3. Results

### 3.1. The Regulation of pH 

The gluconic acid generated during glucose oxidation reduced the pH of the reaction mixture from 6.6 to below 4.0. Under such a low pH, glucose oxidase is inactive [43]. Two strategies based on CaCO_3_ and 1 M NaOH were proposed to avoid unwanted pH changes. Among the various amount of CaCO_3_, the mass corresponded to 4 g/L (the mass introduced dissolves as the reaction proceeds) was selected as the most promising. In the presence of CaCO_3_, the gluconic acid was converted to calcium gluconate, according to Equation (3):2 C_6_H_12_O_7_ + CaCO_3_ → C_12_H_22_CaO_14_ + CO_2_ + H_2_O(3)

The comparison of pH changes during glucose oxidation performed in various variants are presented in Table 1.

The satisfactory buffering properties of CaCO_3_ allow one to maintain the almost constant pH of the reaction mixture during the almost total glucose decomposition. From a practical point of view, the reaction mixture’s initial turbidity resulted from CaCO_3_ disappeared when the high conversion yield was achieved (>75%). In the case of 1 M NaOH, its addition caused a slight increase in the volume of the reaction mixture and dilution of reagents. Nevertheless, both procedures can be applied. 

### 3.2. The Aeration Impact

According to the enzyme classification, the reaction of glucose decomposition with glucose oxidase is based on glucose oxidation. To select the appropriate aeration variants, three strategies were tested: mixing, catalase addition, and exogenous aeration with compressed air (Table 2). 

Based on the obtained results, aeration by mixing or generated by catalase was insufficient, and the reaction was performed under a deficiency of one of the substrates, oxygen. The experiments monitored with an oxygen electrode showed that the addition of glucose oxidase in the first strategy resulted in a rapid lowering of the oxygen concentration, to a value lower than 1 mg/L. While oxygen generated by catalase during the one-pot biocatalysis was quickly consumed by glucose oxidase. In this case, the oxygen concentration was reduced from 10.45 mg/L to 3.36 mg/L immediately after the addition of glucose oxidase. After the first few minutes, it was reduced to a value lower than 1 mg/L. The addition of catalase caused the short-term growth of the oxygen concentration to 5.87 g/L, which was below 0.5 mg/L after a few minutes. Based on the presented results, an exogenous oxygen supplementation was necessary. Comparing all variants after 3 h of glucose oxidation, the conversion yield was evaluated as 13.3%, 18,48%, and 81.9%, for the strategy of mixing, catalase addition, and compressed air, respectively. Continuous oxygen supplementation reduced the reaction time to less than 5 h and allowed complete glucose decomposition. In this case, the oxygen concentration persisted above 20 mg/L.

### 3.3. Catalase Impact

The creation of an enzymatic cascade is justified when all its elements act synergistically. In the case of the cooperation of glucose oxidase and catalase, catalase decomposed hydrogen peroxide, which was generated by glucose oxidase and acted as its inhibitor. Benefits related to catalase addition are presented in Figure 1. 

By the elimination of product (H_2_O_2_) inhibition, catalase improved the yield of glucose decomposition and allowed complete glucose conversion.

### 3.4. Two-Enzymatic Cascade with Native Glucose Oxidase and Catalase 

The efficiency of the enzymatic cascade, regardless of its complexity, depends on its components’ activity. Thus, the first experiments were performed in the presence of individually added glucose oxidase and catalase. The conversion of glucose catalyzed by native glucose oxidase was carried out in the presence of glucose 27.5 g/L at 12 °C. The glucose concentration corresponded to the stoichiometry value, which was calculated based on enzymatic hydrolysis of lactose, at a concentration of 55 g/L. The various enzyme concentrations were tested during biocatalysis at 0.2–2 g/L. Furthermore, the stability of glucose oxidase under operational conditions, such as 12 °C and pH 6.6 was monitored. As a result, the native glucose oxidase exhibited a half-life of more than 29 h. The satisfactory stability of glucose oxidase does not correspond to a better reaction yield. Glucose decomposition lasted more than 24 h, and the bioconversion efficiency didn’t exceed 15%. On the contrary, catalase, the second component of the discussed cascade, allows for the efficient decomposition of hydrogen peroxide at 12 °C in much less time, less than 1 h, as shown in recent work [24].

The second step during the creation of the enzymatic cascade is related to the proportions of the enzymes. In this case, various concentrations of enzymes were tested, such as 0.3–8.0 g/L and 0.2–2.4 g/L, respectively, for glucose oxidase and catalase (all variants were presented in the Appendix A). Among them, the most promising option was 6 g/L of glucose oxidase and 1 g/L of catalase, allowing complete glucose conversion at 7.9 h duration (Figure 2).

The results presented in Figure 1 and Figure 2 showed that glucose conversion is effective in a one-pot reaction in the presence of glucose oxidase and catalase. Furthermore, to obtain the shorter bioconversion time with native enzymes, high enzyme concentrations, exogenous aeration with compressed air, and CaCO_3_ addition must have been used.

### 3.5. Co-Encapsulation of Glucose Oxidase and Catalase

The creation of an encapsulated two-enzymatic preparation was preceded by the individual encapsulation of glucose oxidase and catalase in the alginate matrix. In the case of glucose oxidase, the encapsulation was running out with enzyme leakage. On the contrary, catalase encapsulation was characterized by high efficiency, as shown in the previous work [24]. Therefore, we proposed the additional modification of traditional alginate capsules by cross-linking with glutaraldehyde or carbodiimide for the concept of two-enzymatic preparation. 

The degree of enzyme leakage was determined. If there was a leakage of enzymes into the solution, the glucose decomposition occurred not only in the capsules, but also in the solution after the capsules had been removed. In this case, the solution obtained after the separation of the capsules stabilized with carbodiimide exhibited app. two times lower enzyme activity compared to the capsules with glutaraldehyde (Table 3). Based on this, we can conclude that there was better enzyme retention with carbodiimide cross-linking.

Moreover, both variants of capsules with carbodiimide and glutaraldehyde cross-linking were characterized by similar activity under operational conditions. In both cases, complete glucose decomposition lasted about 67 h without exogenous aeration (Figure 3).

Based on the presented results, the subsequent experiments were performed with capsules carbodiimide cross-linking. The operational stability of co-encapsulated enzymes was determined to confirm good enzyme retention. The studies were carried out for two concentrations of enzymes (Figure 4). In both cases, based on the first-order kinetics, the half-life time was calculated at 4.4 h.

To determine the reuse of enzymes, the batch process was conducted. According to the presented results, co-encapsulated glucose oxidase and catalase can be used in 9 cycles, preserving at the end approximately 40% of the initial activity (Figure 5). Each cycle corresponded to almost 97% glucose bioconversion. 

The subsequent experiments were related to the kinetic characterization of co-encapsulated glucose oxidase and catalase. As shown, glucose decomposition catalyzed by co-encapsulated glucose oxidase and catalase at 12 °C runs according to first-order kinetics (Figure 6). Even for much higher glucose concentrations (i.e., >30 g/L), the proportional dependence between substrate concentration and reaction rate was observed.

### 3.6. Glucose Oxidation in Lactose-Free Milk Catalyzed by Co-Encapsulated Glucose Oxidase and Catalase

The natural medium, lactose free-milk produced based on enzymatic lactose hydrolysis, was utilized in the experiment. The lactose decomposition lasted 1 h; after that, the milk was enriched by glucose at a concentration of 26.3 g/L. In previous results, this value was similar to the glucose solution in the HEPES buffer. The comparison of biocatalysis results performed in the natural medium and HEPES buffer is presented in Figure 7.

The results show the analogous scheme of glucose oxidation in the buffer and lactose-free milk. Apart from that, the earlier proposed aeration and pH regulation strategies also were suitable for milk. In the last case, the NaOH solution proved unquestionably better. The solubility of CaCO_3_, due to the presence in the milk of calcium ions at a high level (1.2 g/L) and other cations and anions, was slowed down, despite the reaction run. As a result, the solid of CaCO_3_ dissolved more slowly and maintaining the pH at the set level must have resulted in a lower reaction rate (using less enzyme preparation) and, consequently, a longer process of the total decomposition of the glucose used.

After biocatalysis, the obtained lactose-free milk was noticeably less sweet (organoleptic measurement). Moreover, its color, texture, and aroma were preserved (organoleptic measurement). The most visible difference between obtained non-sweet milk and initial lactose-free milk was related with milk texture. The non-sweet lactose-free milk was more fluffy due to the aeration conditions. 

## 4. Discussion

The presented work is focused on the two-enzymatic cascade with glucose oxidase and catalase for low-temperature biocatalysis, which is utilized in dairy processes. The two-enzymatic preparation is used for the glucose decomposition from lactose-free milk to reduce its sweetness. Surprisingly, in the proposed method, only catalase was previously described as the enzyme with low-temperature activity [44,45]. In the case of glucose oxidase isolated from *Aspergillus niger*, most literature reports suggest its optimal temperature range of 25–40 °C, and the pH requirements are close to 5.5 [46,47,48,49,50]. Furthermore, the literature reports that food-related biocatalysis based on glucose oxidase was conducted at temperatures above 25 °C [51,52,53].

Experiments carried out with single enzymes showed that glucose oxidase is active at 12 °C, but the complete glucose decomposition was hindered and extended over time. During glucose bioconversion, the problem of pH regulation and oxygen supplementation was analyzed. The phenomenon of pH decreasing during biocatalysis with glucose oxidase was expected in the context of the enzymatic production of gluconic acid. The literature problem-solving proposals were related to the continuous supplementation with strong bases, like NaOH or KOH [11,54,55]. The CaCO_3_ suggestion presented in this work was also previously proposed [55,56]. The CaCO_3_ application, besides buffering properties, brings the benefits related to the production of calcium gluconate, which is used in treating hypocalcemia, cardiac arrest, and cardiotoxicity and protects against osteoporosis [57]. However, our results showed that NaOH regulation is better at a high reaction rate. The solubility of CaCO_3_, due to the presence in the milk of calcium ions at a high level (1.2 g/L) and other cations and anions, was slowed down. As a result, the solid of CaCO_3_ dissolved more slowly and maintaining the pH at the set level must have resulted in a lower reaction rate (using less enzyme preparation) and, consequently, a longer process of the total decopomposition of the glucose used.

The presented studies show that the oxygen concentration decreases as soon as glucose oxidase is added to the reaction mixture. Furthermore, the addition of catalase slightly increased the oxygen concentration. Finally, after the first few minutes of biocatalysis, the amount of oxygen in the reaction mixture was exhausted in both cases without and with catalase. The obtained results are similar to those presented by Tao et al. [58]. The literature reports, which describe the activity of native or immobilized glucose oxidase, do not always include oxygen supplementation. Some don’t give information about the necessity of aeration [47,59,60] while others suggest an intensive aeration of the reaction mixture before biocatalysis starts [15,61]. In some papers, there is a recommendation for catalase addition and the utilization of its enzymatic activity to oxygen production [58]. Fortunately, there is also information about continuous aeration [62], utilization of bubble columns [63], or air-lift reactors [64]. 

Our results show that the oxygen concentration resulting from catalase activity, which decomposes hydrogen peroxide at a concentration corresponding to the stoichiometry of 27.5 g/L glucose decomposition, is insufficient to obtain the high yield and quick time of glucose bioconversion. Even high glucose oxidase concentrations don’t shorten the reaction time, which was more than 24 h. Continuous external aeration brought the expected effect. The conversion of glucose at an initial concentration of 27.5 g/L lasted less than 5 h, using simultaneously much lower enzymes concentrations, in contrast to the conditions without aeration. Additionally, the benefits related to the presence of catalase in the reaction medium and thus elimination of product inhibition were more visible. In the case of catalase, the enzyme exhibited satisfactory activity under the analyzed conditions. It allows the rapid decomposition of H_2_O_2_ at 12 °C. 

A co-encapsulation strategy inside the alginate matrix was proposed to increase the usefulness of the proposed two-enzymatic cascade in food applications. It is important to note that during encapsulation, the most crucial parameter is the molecular mass of the enclosed enzymes. In the presented case, glucose oxidase and catalase are characterized by various molecular masses, respectively, 180 and 240 kDa. These differences were especially visible during individual encapsulation. Preparations based on encapsulated catalase showed 100% enzyme retention in the alginate matrix, as shown in the previous publication [44] and by other authors [65]. This result has not been confirmed for glucose oxidase. In this case, significant losses of enzyme over time (approximately 10–12% per cycle) were observed. Thus, the creation of co-encapsulated glucose oxidase and catalase inside the traditional alginate matrix was unprofitable. For comparison, examples of effective glucose oxidase encapsulation in alginate beads can be found in the literature [66]. Surprisingly, Blandino and co-workers presented a high retention of glucose oxidase in the alginate matrix in the presence of 1% (*w*/*v*) sodium alginate and 5.5% (*w*/*v*) CaCl_2_. Our results showed high enzyme leakage even at 1.5% (*w*/*v*) sodium alginate and 15% (*w*/*v*) CaCl_2_. For the effective encapsulation of glucose oxidase, the solutions related to the combination of alginate with chitosan [67], hollow capsules with carboxymethyl cellulose [60] as well as alginate microspheres prepared by the method of emulsion-conjugation [68], were proposed.

Emerging difficulties with individual glucose oxidase encapsulation affected the method of creation of the co-encapsulated glucose oxidase and catalase in the alginate matrix. In this case, the classic alginate beads were enriched by cross-linking with glutaraldehyde or carbodiimide. As shown, cross-linking with carbodiimide was characterized by good enzyme retention. Furthermore, there are no significant differences between the capsules with glutaraldehyde or carbodiimide in the context of enzymatic activity. Utilising glutaraldehyde and carbodiimide during glucose oxidase encapsulation is a new proposal. This strategy allows for the complete enzymatic preparation that meets all the requirements of food biocatalysis.

Moreover, the properties of co-encapsulated glucose oxidase and catalase, their stability, relative high half-life time, and the possibility of reuse allowed the process to be conducted in batch mode. Additionally, the kinetic characteristics of co-encapsulated glucose oxidase and catalase can be described by first-order kinetics. Surprisingly, the individually encapsulated catalase showed the classic Michaelis-Menten kinetics, as shown in the previous work [44]. Furthermore, in the case of immobilized glucose oxidase, most of the literature reports also suggested the kinetic of Michaelis-Menten [62,66]. The Michaelis-Menten scenario was also confirmed for co-immobilized glucose oxidase and catalase [16,37].

## 5. Conclusions

The popularity of biocatalysis is still growing. Today, enzyme-based green technologies have become an inseparable element of food processing. They not only improve taste and food quality, but they also can be involved in the fight against obesity by reducing the calorific value of food via glucose decomposition. The current requirements of industrial biocatalysis are related to enzyme co-immobilization. The challenge is to create a co-immobilized preparation and ensure the appropriate operational conditions for all biocatalysts. Due to the growing demand for lactose-free products, the presented work focused on the enzymatic glucose decomposition by a two-enzymatic cascade operating at 12 °C to reduce the sweetness of lactose-free products. The simplified flowsheet to achieve the presented assumption is presented in Figure 8.

The presented results suggest that ‘the bottleneck’ of the one-pot cascade based on glucose oxidase and catalase worked at 12 °C is related to glucose oxidase. Benefits related to the presence of catalase refer to a reduction of product (H_2_O_2_) inhibition. Still, the amount of oxygen generated by a catalase is insufficient to maintain the high activity of glucose oxidase. Continuous exogenous aeration is indispensable in completing glucose decomposition in a relative short time at 12 °C. The glucose decomposition, due to the temperature of bioconversion as well as the possibility of batch mode realization, can be performed during milk storage and transportation. The utility properties are also increased by the preparation size (dimeter close 4 mm) and shape (microcapsules), ensuring their easy separation. 

## Figures and Tables

**Figure 1 foods-12-00113-f001:**
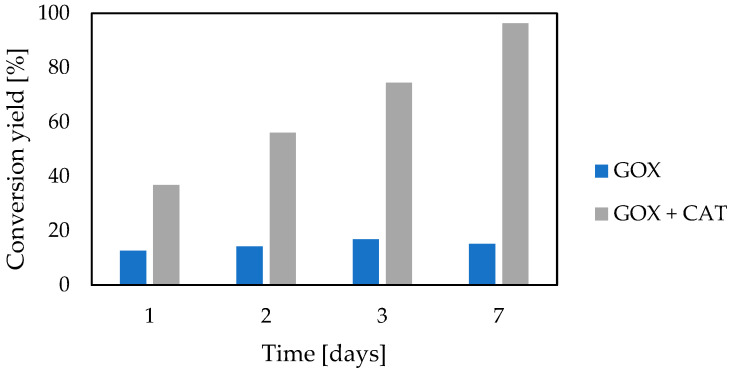
Impact of native catalase on the efficiency of glucose decomposition by native glucose oxidase (glucose 27.5 g/L, glucose oxidase 0.15 g/L, catalase 0.4 g/L, 0.1 M HEPES buffer pH 6.6, 12 °C, CaCO_3_ 4 g/L, without exogenous aeration).

**Figure 2 foods-12-00113-f002:**
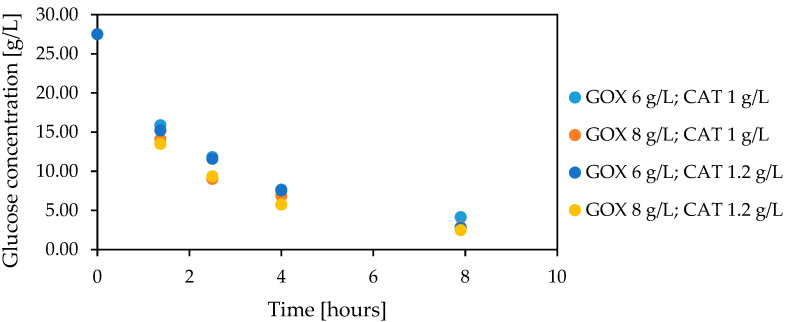
Selection of the proportion of native glucose oxidase and catalase during glucose decomposition (glucose 27.5 g/L, 0.1 M HEPES pH 6.6, 12 °C, CaCO_3_ 4 g/L, without aeration).

**Figure 3 foods-12-00113-f003:**
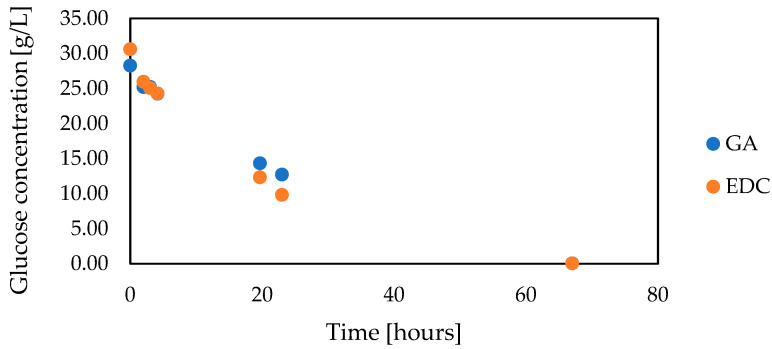
Glucose decomposition by co-encapsulated glucose oxidase and catalase without exogenous aeration (capsules volume 3 mL, total volume 10 mL, glucose oxidase concentration 1.05 g/L, catalase concentration 0.09 g/L, 0.1 M HEPES buffer pH 6.6, 12 °C, CaCO_3_ 4 g/L, without exogenous aeration).

**Figure 4 foods-12-00113-f004:**
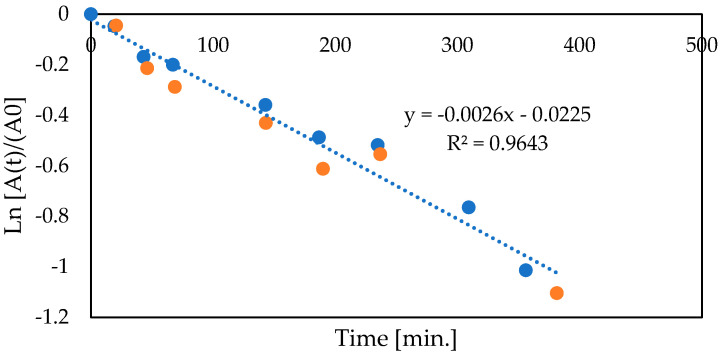
Half-life time of co-encapsulated glucose oxidase and catalase (blue points GOX 1.38 g/L, CAT 0.22 g/L, orange points GOX 2.76 g/L, CAT 0.45 g/L, glucose 27.5 g/L, capsules volume 3 mL, total volume 18 mL, 0.1 M HEPES buffer pH 6.6, 12 °C, CaCO_3_ 4 g/L, compressed air 3 L/h).

**Figure 5 foods-12-00113-f005:**
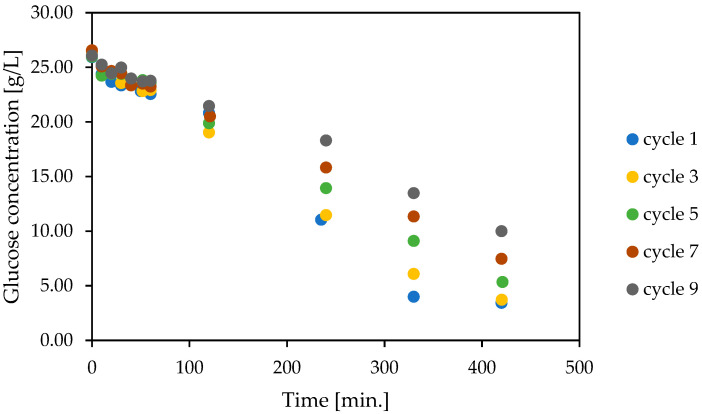
The re-using of co-encapsulated glucose oxidase and catalase, (glucose 27.5 g/L, GOX 1.38 g/L, CAT 0.22 g/L, capsules volume 3 mL, total volume 18 mL, 0.1 M HEPES buffer pH 6.6, 12 °C, CaCO_3_ 4 g/L, compressed air).

**Figure 6 foods-12-00113-f006:**
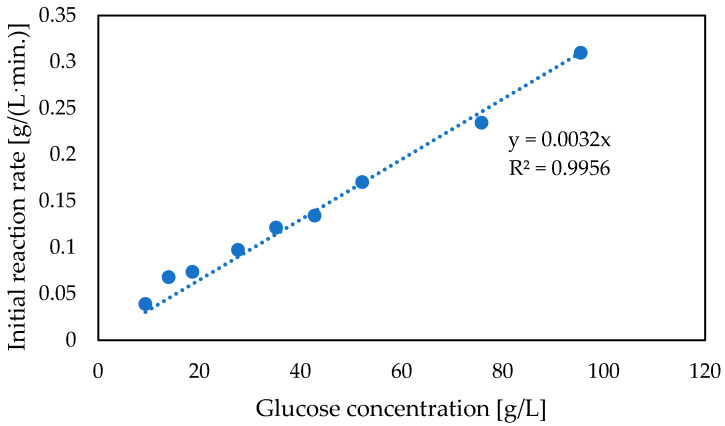
Kinetics of glucose conversion with co-immobilized glucose oxidase and catalase, (GOX 1.33 g/L, CAT 0.22 g/L, capsules volume 3 mL, total volume 18 mL, 0.1 M HEPES buffer pH 6.6, 12 °C, CaCO_3_ 4 g/L, compressed air 3 L/h).

**Figure 7 foods-12-00113-f007:**
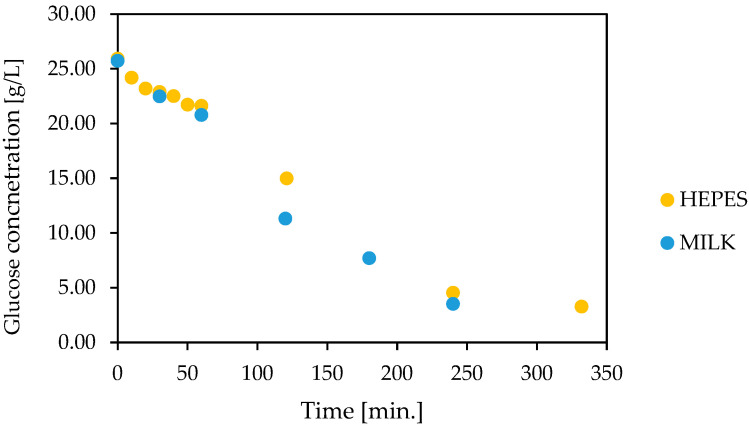
The comparison of glucose oxidase in lactose-free milk and HEPES buffer catalyzed by co-encapsulated glucose oxidase and catalase (GOX 2.7 g/L, CAT 0.43 g/L, capsules volume 3 mL, total volume 18 mL, 0.1 M HEPES pH 6.6, lactose-free milk pH 6.7, 12 °C, 1 M NaOH, compressed air 3 L/h).

**Figure 8 foods-12-00113-f008:**
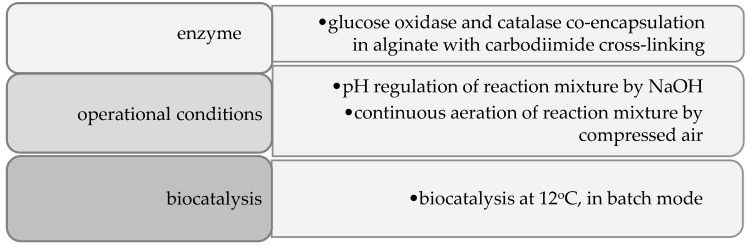
The road map of low-temperature biocatalysis to the reduction of calorific value and sweetness of lactose-free milk.

**Table 1 foods-12-00113-t001:** Changes in pH during glucose oxidation catalyzed by native glucose oxidase (glucose 27.5 g/L, GOX 0.3 g/L, 0.1 M HEPES pH 6.6, 12 °C.

Time [h]	GOX	GOX + NaOH	GOX + CaCO_3_
	pH		
0	6.59	6.65	6.68
1	6.36	6.84	6.61
3	6.0	6.71	6.54
24	3.69	6.62	6.35

**Table 2 foods-12-00113-t002:** The impact of the type of aeration on glucose conversion (α) catalyzed by glucose oxidase (glucose 27.5 g/L, native glucose oxidase 0.3 g/L, CAT 1.0 g/L, 0.1 M HEPES buffer pH 6.6, 4 g/L CaCO_3_, 12 °C).

Type of Aeration	Mixing		Catalase Action		Compressed Air	
Enzymes	GOX		GOX + CAT		GOX + CAT + Aeration	
Time [min]	O_2_ [mg/L]	α [%]	O_2_ [mg/L]	α [%]	O_2_ [mg/L]	α [%]
0	10.48	0	10.45	0	10.60	0
1	0.88	3.3	3.36	6.45	18.61	10.05
15	0.45	4.88	0.70	8.17	>20.0	18.37
60	0.43	13.48	0.64	16.49	>20.0	69.74
180	0.50	15.1	0.56	21.6	>20.0	82.02

**Table 3 foods-12-00113-t003:** The enzymatic activity (a glucose conversion degree) observed in the solution after the capsules removing, (glucose 2.38 g/L, 0.1 M HEPES buffer pH 6,6, 25 °C, CaCO_3_ 4 g/L, without aeration).

Type of Capsules	Alginate	Alginate + GA	Alginate + EDC
Time [h]	Glucose Conversion [%]		
0	0	0	0
5.3	63.84	32.35	16.13
22	98.6	77.31	35.89

## Data Availability

Data is contained within the article or Appendix A.

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
