# Peer review of "Critical Parameters in an Enzymatic Way to Obtain the Unsweet Lactose-Free Milk Using Catalase and Glucose Oxidase Co-Encapsulated into Hydrogel with Chemical Cross-Linking"

_foods, 2022, doi:10.3390/foods12010113_

Round 1

Reviewer 1 Report

Please rephrase the title, it is unclear and not correct (according to me). 

Reference to milk in the title can only be there if authors supply additional experiments showing that their solution also works in milk. There are NO experiments with milk in the article in its current form, so any reference to milk in the title or abstract should be removed.

Health promotion has not been shown in the article, so it may be suggested in the discussion (when experiments with milk are added), but should not be part of the title and also no reference to health aspects (like osteoporosis) should be done in the abstract.

Caloric reduction of milk by enzymatic conversion is also questionable. The authors are invited to substantiate this claim by a clear calculation including all calories present in the treated milk, including fat, protein, sugars and organic acids. Take into account that gluconic acid is only slightly less caloric compared to glucose (3 vs. 4 kcal/g)

Additional remarks:

Table 2 shows oxygen concentration in the different samples but the title mentions glucose conversion. Please adapt. Also, to show the effect of the enzyme it would indeed be more informative to (also) show the glucose conversion in this table.

Par. 3.3 Make better clear that this experiment is about the stability of the enzymes during days of incubation, in contrast to the experiment in par 3.2

Par 3.4 text describes a wide range of different enzyme concentrations that were tested, but the Figure only shows a few selected ones. It is unclear to me why the authors decides to exclude the data on lower enzyme dosage which may be relevant to judge the kinetic properties of the one-pot approach.

The conclusion on lines 256-258 cannot be judged by the reader since data is lacking. The effect of low enzyme concentration, aeration, and/or CaCO3 addition on glucose conversion is not shown by the authors.

Table 3 mentions the measurement of enzymatic leakage. I do not understand this. Is this not a measurement of enzyme inactivation with crosslinkers (GA or EDC)? Please also explain the difference in activity shown in Table 3 and Figure 3

Reviewer 2 Report

Title suggested - CRITICAL PARAMETERS IN AN ENZYMATIC WAY TO OBTAIN THE HEALTH PROMOTING – UNSWEET, LOW-CALORIC AND CALCIUM ION ENRICHED (WRITE FURTHER – INGREDIENT OR PRODUCT)

Kindly write the full-form of several abbreviations once when they are mentioned for the first time viz.,

DNS, HEPES, PBS, MES, CAT, EDC,

Abstract

With respect to calcium enrichment of material, mention specifically that since CaCO3 was used for pH regulation, such treatment led to calcium enrichment of the material. (or else I was under impression that along with use of 2 enzymes viz., catalase and Tglucose oxidase, calcium ions were added for enrichment)

Introduction

Butter hardly contains any lactose – so need not mention butter as a product from where lactose is to be eliminated. Even hard cheese varieties are very low in lactose content – same comment prevails for cheese too.

Page 2, line 62 – preservation of nutritive value (21) – which nutrient is preserved?

Page 2, line 74 – which chemical was used for crosslinking purpose; and for which specific enzyme?

Page 2, line 93 – Which innovative method was used for encapsulation – so that it is mentioned ‘innovative encapsulation strategy’

Page 2, line 96 – Relative sweetness is better term than sweetness factor

Materials and Methods

Page 3, Line 120 - Name the type of Spectrophotometer used with its place of manufacture

Page 3, Line 124  - Any measurement of pressure of compressed air?

Page 3, Line 133 – After how much time of start of enzymic action using blend of 2 enzymes was CaCO3 added; was it again added after some interval to maintain the pH (what was that specific pH that was attempted to maintain?)

Page 3, Line 143 – Specify as Appendix table – Glucose oxidase 8 g/L; how much Catalase concentration was used (i.e. range only has been specified by the authors viz., 8-16 g/L for Glucose oxidase and 1.3-2.6 g/L for Catalase – specify the exact combination concentration)

Page 4, Line 149 – Which was the crosslinking agent used here?

Page 4, Line 157 – Which was the device used for stirring (provide its specification too)

Page 4, Line 173 – What was the air flow parameters?

Page 4, Line 177 – What was the significance level tested – 5% or 1% i.e. mention (p<0.05) when writing ‘significantly different’; which statistical tool was used – Provide Table to show the Critical Difference values (with Sem and Coefficient of Variation too)

Page 5, Line 190 – Avoid word ‘Superb’ in scientific literature; use other appropriate word

Page 5, Line 191 – In entire text, 100% glucose was never decomposed (only 98% or so) – avoid printing ‘total glucose decomposition’

Page 5, Line 193 – What was the maximum conversion yield? – specify

Page 6, Figure 1 – Mention on right hand side – Glucose oxidase plus Catalase (instead of printed – ‘Catalase’

Page 6, Line 229 – Why word ‘native’ for enzyme is required?? Is it highly essential?

Page 6, Line 238 – 55 g/L is concentration of which component? Unable to comprehend

Page 6, Line 248 – Specify combination concentration of both enzymes individually i.e. Glucose oxidase at 0.3 g/L along with Catalase of what concentration? (Only range has been mentioned)

Page 6, Line 251 – Instead of ‘after 7.9 hours’ it should be printed ‘at 7.9 hours duration’

Page 7, Line 271 - Here according to my understanding - 'glutaraldehyde' cross-linking should be specified - earlier wrote carbodiimide exhibited lower enzyme activity.... check it out??

Page 10, Line 326 – CaCO3 is not a strong base at all – as mentioned in this line.

Page 10, Line 338 – Intensive Medium aeration – ‘Intensive’ and ‘medium’ do not go well together

Page 10, Line 339 – recommendation for catalase – statement seems incomplete

Page 10, Line 342 – generated, thanks to…   seems not much suitable wordings

Page 10, Line 359 – What is meant by word ‘high’ (specify value) for encapsulated catalase

Page 10, Line 361 – What is meant by word ‘high’ (specify value) for enzyme leakage from capsulated form

Page 10, Line 364 – The author presented – who is the author being referred to here?

Page 11, Line 373 - Alginate beads got enriched – enriched with which component?

Page 11, Line 397 – ‘operating at’ would be more suited words than ‘acting at’

Page 11, Line 411 – What is meant by ‘preparation size’?

Under Conclusion

Specify how much calorie reduction was feasible through use of the blend of 2 enzyme activity; how much calorie did the control (unhydrolyzed material) sample confer?

References

Rizzo, P.V. (V is in small case; should be made capital letter)

Czyzewska, K. and Trusek, A. (2021) - Words Vol. and page should not appear (These aspects viz., vol. and page nos. have been repeated there also!)

Bie, J. et al. (2022) – Mention publisher name and place of publication (Editor names too)

Liu et al. (2022) – Abbreviate the name of the Journal (as per Journals format)

Agyei, D. et al. (1984) - Based on Title and Italicized topic - looks like Book chapter reference - No mention of name of Book, Publisher, name of place (Country), etc.

Blandino et al. (2001) – Print abbreviated name as ‘Process Biochem’

Milek J (2011) – Print Aspergillus niger; Print name of author as ‘Milek J’

Singh and Verma (2013) Print Aspergillus niger in Italics style. Ref. No. 47, 48 – Do same aspect.

Lanyan et al. (2020) First alphabet of each word in title of paper to be in capital letter.

Karmali et al. (2004) Print both microorganism name in Italics – Aspergillus niger and Penicillium amagasakiense

Blandino et al. (2001) Print abbreviated name of Journal ‘Biochem.’

Adanyi et al. (1999) and Wingard et al. (1984) - Abbreviate the Journal name properly; 1984 9:1 (is repeated in second Ref. Wingard et al. (1984))

Mafra et al. (2019) and Jonovic et al. (2021) - Volume number and page numbers are repeated in Reference

Colak et al. (2012) and Agyei et al. (1984) - Seems like Book Reference; printed as Journal Reference – do check the same

Tao et al. (2009). Print proper Journal name – Biophys J.

Chakraborty and Can (2022) Print Publisher, place of publication, name(s) of editor(s) of this Book

Ref. No. 33 – Name of Journal is entirely missing

Ref. No. 45 – ‘2’ of H2O2 to be printed as subscript

Ref. No. 55, 56 – both patents – Write Patent numbers and provide website (United spelling incorrectly printed)
